# Bactericidal Activity of Carvacrol against *Streptococcus pyogenes* Involves Alteration of Membrane Fluidity and Integrity through Interaction with Membrane Phospholipids

**DOI:** 10.3390/pharmaceutics14101992

**Published:** 2022-09-21

**Authors:** Niluni M. Wijesundara, Song F. Lee, Zhenyu Cheng, Ross Davidson, David N. Langelaan, H. P. Vasantha Rupasinghe

**Affiliations:** 1Department of Biology, Faculty of Science, Dalhousie University, Halifax, NS B3H 3E2, Canada; 2Department of Plant, Food, and Environmental Sciences, Faculty of Agriculture, Dalhousie University, Truro, NS B2N 5E3, Canada; 3Department of Animal Science, Faculty of Animal Science and Export Agriculture, Uva Wellassa University, Badulla 90000, Sri Lanka; 4Department of Microbiology & Immunology, Faculty of Medicine, Dalhousie University, Halifax, NS B3H 3E2, Canada; 5Department of Applied Oral Sciences, Faculty of Dentistry, Dalhousie University, Halifax, NS B3H 3E2, Canada; 6Canadian Center for Vaccinology, Dalhousie University, Nova Scotia Health Authority, and the Izaak Walton Killam Health Centre, Halifax, NS B3H 3E2, Canada; 7Department of Pathology, Faculty of Medicine, Dalhousie University, Halifax, NS B3H 3E2, Canada; 8Laboratory Medicine, Division of Microbiology at the Queen Elizabeth II Health Sciences Centre, Nova Scotia Health Authority, Halifax, NS B3H 1V8, Canada; 9Department of Biochemistry & Molecular Biology, Faculty of Medicine, Dalhousie University, Halifax, NS B3H 3E2, Canada

**Keywords:** carvacrol, streptococcal pharyngitis, *Streptococcus pyogenes*, mechanism of action, membrane, phospholipids, permeability

## Abstract

Background: Carvacrol, a mono-terpenoid phenol found in herbs, such as oregano and thyme, has excellent antibacterial properties against *Streptococcus pyogenes*. However, its mechanism of bactericidal activity on *S. pyogenes* has not been elucidated. Objectives: This study investigated the bactericidal mechanism of carvacrol using three strains of *S. pyogenes*. Methods: Flow cytometry (FCM) experiments were conducted to determine carvacrol’s membrane permeabilization and cytoplasmic membrane depolarization activities. Protoplasts of *S. pyogenes* were used to investigate carvacrol’s effects on the membrane, followed by gel electrophoresis. The carvacrol-treated protoplasts were examined by transmission electron microscopy (TEM) to observe ultrastructural morphological changes. The fluidity of the cell membrane was measured by steady-state fluorescence anisotropy. Thin-layer chromatographic (TLC) profiling was conducted to study the affinity of carvacrol for membrane phospholipids. Results: Increased membrane permeability and decreased membrane potential from FCM and electron microscopy observations revealed that carvacrol killed the bacteria primarily by disrupting membrane integrity, leading to whole-cell lysis. Ultra-structural morphological changes in the membrane induced by carvacrol over a short period were confirmed using the *S. pyogenes* protoplast and membrane isolate models in vitro. In addition, changes in the other biophysical properties of the bacterial membrane, including concentration- and time-dependent increased fluidity, were observed. TLC experiments showed that carvacrol preferentially interacts with membrane phosphatidylglycerol (P.G.), phosphatidylethanolamine (P.E.), and cardiolipins (CL). Conclusions: Carvacrol exhibited rapid bactericidal action against *S. pyogenes* by disrupting the bacterial membrane and increasing permeability, possibly due to affinity with specific membrane phospholipids, such as P.E., P.G., and CL. Therefore, the bactericidal concentration of carvacrol (250 µg/mL) could be used to develop safe and efficacious natural health products for managing streptococcal pharyngitis or therapeutic applications.

## 1. Introduction

Carvacrol, 2-methyl-5-(propan-2-yl) phenol, is a monoterpene phenolic compound that is found in various herbal plants belonging to the genera *Thymus* and *Oregano* [1,2,3,4]. Carvacrol inhibits the growth of many Gram-positive and Gram-negative pathogenic bacteria, including methicillin-resistant *Staphylococcus aureus, Streptococcus pneumoniae*, and *Haemophilus influenzae,* which cause various upper respiratory tract infections [5,6].

We have recently reported the antibacterial activity of carvacrol against *Streptococcus pyogenes* [7], a major upper respiratory tract pathogen that causes diseases with significant morbidity and mortality worldwide [8,9]. One of the common diseases caused by *S. pyogenes* is streptococcal pharyngitis, which has primarily been treated with penicillin without significant buildup of resistance over the decades [10,11]. However, cases caused by some erythromycin-resistant *S. pyogenes* strains have been identified [12,13]. According to our previous findings, carvacrol showed effectiveness against an erythromycin-resistant strain of *S. pyogenes,* spy 1558, at a minimum inhibitory concentration (MIC) of 125 µg/mL [7]. Therefore, we have initiated an understanding of carvacrol’s potential mode of action against this bacterium.

Furthermore, cellular content leakage and ultra-structural modification were observed upon carvacrol treatment, suggesting that carvacrol may have affected the cell wall and/or cell membrane [7]. Therefore, carvacrol has the potential to be a promising alternative to antibiotics to prevent, treat, and manage streptococcal pharyngitis. Moreover, carvacrol is a pain-relieving natural health product; therefore, we need to properly understand how carvacrol works as an antibacterial agent at the cellular level. Several studies have investigated the impacts of carvacrol on Gram-negative bacteria. For example, carvacrol disrupts the cell membrane, inhibits protein biosynthesis, and interrupts the pathogenic life cycle, including adhesion and biofilm formation [2,14,15,16]. However, exact bactericidal mechanisms of carvacrol on *S. pyogenes* have not been reported. In the current study, we investigated the effects of carvacrol on the changes in membrane permeabilization, membrane potential, and membrane fluidity using intact *S. pyogenes* cells and protoplasts.

## 2. Materials and Methods

### 2.1. Media and Chemicals

Brain heart infusion (BHI) was purchased from Oxoid Ltd. (Nepean, ON, Canada) and prepared according to the manufacturers’ specifications. Carvacrol was purchased from Sigma-Aldrich Ltd. (Oakville, ON, Canada) and was maintained at 4 °C. Sodium chloride (NaCl ≥ 99.0%), sodium acetate (NaOAc), and magnesium chloride (MgCl_2_) were purchased from BioShop (BioShop^TM^ Canada Inc., Burlington, ON, Canada). Casamino acids (Bacto™, Dickinson and Company, Baltimore, MD, USA), glucose (EM Science, Gibbstown, NJ, USA), and phenylmethanesulphonyl fluoride (PMSF) (BDH chemicals Inc., ON, Canada) were used for the buffer preparations. Agarose (Thermo Fisher Scientific, Waltham, MA, USA), vanillin, penicillin G sodium salt, daptomycin, Dulbecco’s phosphate-buffered saline (PBS), dimethyl sulfoxide (DMSO) (≥99.8%), ethidium bromide (3,8-diamino-5-ethyl-6-phenylphenanthridinium bromide, >95%), sucrose (1-O-α-D-glucopyranosyl-β-D-fructo-furanoside), and mutanolysin were obtained from Sigma-Aldrich Ltd. (Oakville, ON, Canada). L-α-phosphatidyl-DL-glycerol sodium salt (P.G.), 1-palmitoyl-2-oleoyl-sn-glycero-3-phosphoethanolamine (P.E.), L-α-phosphatidylcholine (P.C.), cardiolipin sodium salt (CL), and cholesterol (Ch) were also purchased from Sigma-Aldrich Ltd. (Oakville, ON, Canada).

### 2.2. Bacterial Strains and Growth Conditions

Three *S. pyogenes* strains, ATCC 19615, a clinical isolate (isolated from a pharyngitis patient), and an erythromycin-resistant strain (spy 1558, erm), were used in the study and cultured in BHI at 37 °C and 5% CO_2_. Stock cultures were stored at −80 °C in BHI containing 20% glycerol. When required, the subcultures were made on BHI agar plates and grew in a humid 5% CO_2_ incubator (Model 3074, VWR International, West Chester, PA, USA) at 37 °C. Subcultures were refreshed every 2 weeks. The inoculum suspension in the broth was prepared and diluted to 1 × 10^6^ CFU/mL (OD_600_ = 0.02) in the BHI medium, as described previously [1].

### 2.3. Membrane Permeability

Cell membrane permeability following carvacrol treatments was assessed by flow cytometry using LIVE/DEAD BacLight™ Bacterial Viability Kit (L7012, Molecular Probes Inc, Eugene, OR, USA). The bacteria were stained with SYTO9, a membrane-permeable green-fluorescent nucleic acid stain or propidium iodide (PI), a membrane-impermeable red-fluorescent nucleic acid stain.

*Test sample preparation*: *S. pyogenes* cells from cultures in the mid-exponential growth phase were harvested by centrifugation at 10,000× *g* for 10 min, washed, and resuspended to an OD_600_ = 0.02. Then, cell suspensions were diluted to 1 × 10^6^ CFU/mL in the phosphate-buffered saline (PBS). Next, the cells were exposed to carvacrol at the concentrations of 2 × MIC, 1 × MIC, 1/2 × MIC, or DMSO vehicle control for different periods (30 min, 1 h, 16 h, and 24 h) at 37 °C at 5% CO_2_. The cells were stained, as described below.

*PI staining:* PI (3 µL, 20 mM) was incubated with 1 mL of cells for 20 min in the dark. A dead cell control (membrane-compromised cells) was prepared from an overnight culture, and the cells were heat-killed at 95 °C for 15 min. The dead cell control (1 × 10^6^ CFU/mL) was similarly stained. The samples were analyzed by flow cytometry within 1 h of staining.

*SYTO9 staining:* A viable cell sample was prepared from an overnight culture not treated with carvacrol and served as the 100% live-cell control. The test cell sample and viable cell control (1 × 10^6^ CFU/mL) were incubated with 3 µL of 3.34 mM SYTO9 in DMSO at room temperature for 15 min. The samples were maintained in the dark at room temperature for no more than 1 h and analyzed by flow cytometry.

*Double-staining with PI and SYTO9:* Carvacrol- or DMSO-treated cell suspensions were diluted to reach a final density of 1 × 10^6^ CFU/mL bacteria (OD _600_ = 0.02). A mixture of stains was prepared by combining PI (20 mM) and SYTO9 (3.34 mM) (1:1 ratio), and 3 μL of the combined stain mixture was added to each of the samples (treated samples and DMSO vehicle control sample). Then, the tubes were incubated at room temperature in the dark for 20 min before measuring the fluorescence of bacterial suspensions with a fluorescence-activated cell sorter (FACS) flow cytometer (CytoFLEX, Beckman Coulter Inc., Indianapolis, IN, USA).

*Flow cytometry (FCM):* To measure the fluorescence intensity, the laser excitation/emission wavelength of 485/542 nm for SYTO9 and 485/610 nm for PI were used. Background fluorescence from the medium was determined, and the results were corrected as necessary. Cell suspensions without the carvacrol treatment served as the control, and 10,000 events were recorded for each sample. Data acquisition was controlled by CytExpert^TM^ software (version 2.1, Beckman Coulter Inc., Indianapolis, IN, USA) and analyzed using the FACS express software (version 5, De Novo Software, Glendale, CA, USA).

### 2.4. Fluorescence Microscopy of Bacterial Viability

Microscopic comparison of live/dead cells was performed using a LIVE/DEAD BacLight Bacterial Viability Kit (L7007, Molecular Probes Inc., Eugene, OR, USA). Briefly, late exponential phase bacteria (ATCC 19615) resuspended to 1 × 10^6^ cells/mL were treated with carvacrol for 16 h at 37 °C. Samples treated with DMSO were included as controls. Equal volumes of PI and SYTO9 were combined, and 3 µL of the dye mixture was added to 1 mL of the bacterial suspension and incubated at room temperature for 15 min in the dark. A 5 µL of the stained bacterial suspension was examined under a fluorescence microscope (Axio Imager 200 M, ZEISS, Gottingen, Germany) equipped with a 63× magnification oil immersion objective.

### 2.5. Membrane Potential

The effects of carvacrol on the membrane potential/cytoplasmic membrane depolarization activity were determined according to the BacLight™ Membrane Potential Kit instructions (B34950, Molecular Probes Inc, Eugene, OR, USA).

*Sample preparation*: Five-milliliter samples of *S. pyogenes* ATCC 19615 (1 × 10^8^ CFU/mL) were treated with carvacrol (2 × MIC, 1 × MIC, 1/2 × MIC; MIC = 125 µg/mL) or DMSO vehicle control. Ten microliters of 500 µM of carbonyl cyanide 3-chlorophenylhydrazone (CCCP), a proton ionophore, were added to 1 mL of each sample. The samples were incubated at 37 °C at 5% CO_2_. At 30 min, 1 h, 16 h, and 24 h, the cells were harvested and diluted in BHI to approximately 1 × 10^6^ CFU/mL. The cells (1 mL) were stained with 10 µL of 3 mM of 3, 3′-diethyloxacarbocyanine iodide [DiOC2(3) (Molecular Probes, Eugene, OR, USA)], a membrane potential-sensitive fluorescent probe, for 30 min at 37 °C.

*FCM:* Flow cytometry was performed at excitation and emission wavelengths of 622 and 670 nm, respectively. Background fluorescence resulting from the medium was determined. Ten thousand events were recorded for each sample. The data were expressed by mean fluorescence intensity (MFI).

### 2.6. Protoplast Experiments

#### 2.6.1. Preparation of Protoplasts

Protoplasts were cultivated according to the methods described by Linder et al. [17] and Parks et al. [18] with modifications. Bacterial culture was harvested at the mid-exponential phase, adjusted to OD_600_ = 0.6, and centrifuged for 5 min at 10,000× *g*. The cells were washed once in a sterilized protoplast wash buffer (50 mM sodium acetate pH 6.5, 0.2 mM MgCl_2_), and were re-suspended in 200 μL warmed (37 °C) protoplast buffer (50 mM sodium acetate, 0.2 mM MgCl_2_, 30% sucrose, 0.1% glucose, 0.05% casamino acids, pH 6.5) containing 0.5 mM phenylmethylsulfonyl fluoride (PMSF) and 800 units of mutanolysin. Samples were incubated at 37 °C for 30 min and were then used to inoculate 2 mL of BHI media containing 30% of sucrose and 0.06 μg/mL of penicillin G. Protoplast cultures were incubated for 20 h at 37 °C at 5% CO_2_. All solutions and media were filtered, sterilized, and stored at room temperature.

#### 2.6.2. Confirmation of Protoplast Formation

##### Microscopy of Protoplast Samples

A sample obtained from overnight protoplast culture was examined under a light microscope to confirm that protoplasts had formed. Next, the osmotic support of the protoplast sample was removed by adding water, and burst-open cells were observed under light microscopy. Briefly, the pellet was collected after 1 mL of the overnight culture was centrifuged (10,000× *g* for 5 min). A sample drop on a glass slide was examined under light microscopy after the pellet had been redissolved in 100 µL of water.

Further verification of the proper formation of protoplasts was accomplished by observing the protoplast cultures via transmission electron microscopy (TEM). Briefly, an overnight protoplast culture (1 mL) was centrifuged (10,000× *g* for 5 min), and the pellet was washed once in the protoplast buffer. Then, as described previously [4], the pellet was fixed, dehydrated, embedded in resin, and visualized in thin-stained sections.

##### Gel Electrophoresis of Protoplast Samples

The 1 mL protoplast culture of *S. pyogenes* was centrifuged (10,000× *g* for 5 min), and the pellets were collected. Then, 0.8% agarose gel electrophoresis was performed for the pellets before and after adding water (100 µL) for leakage of nucleic acids.

#### 2.6.3. Confirmation of the Effect of Carvacrol on Bacterial Membrane

##### Effect of Carvacrol on Protoplast Samples

The protoplast of *S. pyogenes* (ATCC 19615) was prepared and cultivated in 10 mL of BHI media containing 30% of sucrose and 0.06 μg/mL of penicillin G (20 h at 37 °C). Bacteria density was adjusted to OD_600_ = 0.6 and then 2 mL of each bacterial suspension was treated with carvacrol concentrations as 4 × MIC (15 μg/mL), 2 × MIC (7.5 μg/mL), 1 × MIC (3.75 μg/mL), and 1/2 × MIC (1.88 μg/mL) calculated according to the cell density. Protoplasts treated with carvacrol were incubated for 1 h at 37 °C in the presence of 5% CO_2_. Then, TCA perceptible nucleic acid released into the supernatant was determined similarly to the previous method [7].

##### Transmission Electron Microscopy (TEM) of Carvacrol-Treated Protoplast Samples

Samples of each concentration of carvacrol treatment with protoplast were centrifuged (10,000× *g* for 5 min), and the pellets were washed in protoplast buffer. Samples were fixed and visualized using TEM according to the method mentioned in Section 2.6.2. The untreated protoplast was used as a control.

### 2.7. Fluorescence Anisotropy

The changes in the anisotropy values were monitored in the membranes of live bacteria using the 1,6-diphenyl 1,3,5-hexatriene (DPH), a hydrophobic fluorescent probe. Briefly, the mid-exponential *S. pyogenes* (spy 1558) cells were adjusted to an OD_600_ of 0.02. Then, cells were treated with or without carvacrol at 37 °C for 1 and 24 h. Next, cells were centrifuged, washed twice in PBS, and resuspended in PBS. Thereafter, DPH was added to a final concentration of 2 μM and incubated for 10 min at 37 °C. Steady-state fluorescence anisotropy measurements were performed at 37 °C using a PTI QuantaMaster-4-CW spectrofluorometer (Photon Technology International Inc., Birmingham, England). The fluorescence emission was measured at 37 °C using excitation and emission wavelengths of 356 and 423 nm, respectively, and processed using a PTI FeliX32 Analysis module. Fluorescence anisotropy (*r*) was calculated as:$$r=\frac{I_{vv}-GI_{vh}}{I_{vv}+2GI_{vh}}$$
where “*I*” is the fluorescence intensity measured when the excitation and emission polarizers are fixed in the vertical (*v*) or horizontal (*h*) position, the instrument determines “g” to correct for artifacts due to optical components of the fluorimeter.

### 2.8. Thin-Layer Chromatography (TLC) Analysis

Chromatography was performed on 20 × 20 cm silica gel 60 F_254_ glass plates (thickness: 0.5 mm) (Merck Millipore Corporation, Darmstadt, Germany). Phospholipids (P.G., P.E., P.C., CL), cholesterol (Ch), carvacrol, and daptomycin were dissolved in chloroform (25 mg/mL). The individual phospholipid was mixed with carvacrol in chloroform at a ratio of 2:1 (1:1 *v/v*) and incubated for 1 h at 4 °C. Then, 20 volumes of ice-cold acetone were added to the mixture and incubated for 1 h at 4 °C. The precipitated materials were collected by centrifugation (21,000× *g* for 15 min). The supernatant was also saved and dried under a stream of nitrogen gas under the nitrogen evaporator (N-EVAP^TM^, Organomation Association Inc., Berlin, NJ, USA). The precipitate and the dried supernatant samples were dissolved in 20 µL of chloroform. Approximately 5 µL of each sample, pure phospholipids, carvacrol/daptomycin, were loaded on the TLC plates using a chloroform-methanol-acetic acid solvent system (65:25:10 *v/v/v*. The samples were chromatographed for 1 to 1.5 h at room temperature (20 ± 1 °C).

After chromatographic separation, the plates were air-dried for 15 min and sprayed with alcoholic vanillin–sulphuric acid solution (1 g of vanillin, 100 mL 95% ethanol, and 10 mL 95% sulphuric acid). The plates were air-dried for 15 min and then heated in a 100 °C oven until the color spots became visible. The spots were detected under UV using the Bio-Rad^TM^ Gel Doc Imaging System (Bio-Rad Laboratories Inc., MP, Hercules, CA, USA). The Rf values of the compounds were determined as distance moved by the spot front/distance moved by the solvent front. Spots of phospholipids and carvacrol in the precipitates and supernatants and the unbound carvacrol spots in the supernatants were identified by relating the color, spot size, and Rf values to those of the standard phospholipids and carvacrol. Interactions of phospholipids with daptomycin were considered the positive control.

### 2.9. Statistical Analysis

All assays were performed in triplicates, and the results were expressed as the mean ± standard error of three independent experiments. One-way analysis of variance (ANOVA) was performed using Minitab statistical software (version 17.0, Minitab Inc., State College, PA, USA) and GraphPad Prism software (version 5.0, Los Angeles, CA, USA). Statistical differences were defined as (*p* ≤ 0.05), and mean separations among treatments were determined using Tukey’s tests.

## 3. Results

### 3.1. Carvacrol Increases the Permeability of the Bacterial Cell Membrane

To investigate the effects of carvacrol on cell membrane integrity and cell death, we employed PI and SYTO9 staining followed by flow cytometry. *S. pyogenes* treated with carvacrol showed a concentration- and time-dependent increase in the uptake of PI and SYTO9 stains, suggesting that the bacterial cell membrane has been disrupted (Figure 1 and Appendix A). However, this increase was absent in cells not treated with carvacrol. In addition, cells generally showed fewer red fluorescence in the untreated and DMSO controls, indicating that these cells have intact bacterial membranes.

### 3.2. Carvacrol Shows a Concentration-Dependent Increment in Dead Cells

The effects of exposure to bacteriostatic concentrations of carvacrol on cell viability and membrane damage were examined using the LIVE/DEAD BacLight^®^ bacterial viability stains. The results showed that cells treated with 1 × MIC of carvacrol displayed an intense and higher red color than cells treated with DMSO control (Figure 2). Conversely, cells treated with sub-MIC carvacrol levels showed concentration-dependent color-stained cells. Furthermore, the total number of cells decreased with the increasing carvacrol concentrations.

### 3.3. Carvacrol Causes Depolarization of the Cytoplasmic Membrane of S. pyogenes

To reveal whether the antibacterial action of carvacrol involves the disruption of bacterial membrane potential, red and green fluorescence emitted by DiOC2(3) were measured. The red-to-green fluorescence ratio indicates the altered membrane potential compared with the control cells. The scatter plots of green versus red fluorescence exhibited by the bacterial cells are shown in the right panels of Figure 3. Treatment of *S. pyogenes* with the ionophore CCCP resulted in a decreased red-to-green fluorescence ratio, indicating a reduction in membrane potential; however, DMSO control showed a significant reduction only during the early incubation period (Figure 3 and Appendix A). Furthermore, the red-to-green fluorescence intensity ratio dropped significantly with carvacrol over time. Therefore, carvacrol is more potent in membrane depolarization than CCCP. These results suggested that carvacrol may depolarize bacterial cell membrane in a concentration- and time-dependent manner.

### 3.4. Carvacrol-Induced Membrane Damage in Protoplasts

#### 3.4.1. Confirmation of Protoplast Formation

Protoplasts were created by incubating ATCC 19615 and clinical isolate cells with mutanolysin in the presence of sucrose as osmotic support. Protoplasts were cultured overnight in the media containing 30% sucrose and penicillin G to prevent the regeneration of the cell wall. Light microscopic examinations of the culture showed that the cells were rounder in shape and more prominent than intact cells. When the osmotic support was removed, these protoplasts lysed. When the protoplasts were cultured in media lacking penicillin G, the cells were stained, Gram-indicating the regeneration of the cell wall. 

The gel electrophoresis of protoplast supernatants with the removed osmotic support showed nucleic acid leakage in both strains (Figure 4A). The results suggested that the nucleic acids were from protoplast lysis. In addition, TEM analysis showed that the protoplasts have an irregular shape and are more translucent than their intact cells (Figure 4B).

#### 3.4.2. The Effect of Carvacrol on Protoplast Membrane

TEM was used to observe the ultrastructure of *S. pyogenes* protoplasts treated with carvacrol. The untreated protoplasts showed intact membranes and uniformly distributed electron-dense cytoplasms (Figure 4B). In contrast, more detractive morphological features, such as cell membrane disruption, cytoplasmic vacuolations, and cell deformation, were observed in *S. pyogenes* protoplasts treated with >125 μg/mL of carvacrol. At a lower concentration of carvacrol (62.5 μg/mL), damage to the cell membrane was observed, but to a lesser extent (Figure 4B).

### 3.5. Carvacrol Causes Concentration-Dependent Membrane Fluidity Changes

Fluorescence anisotropy was used for studying the rotational diffusion of DPH within the fatty acyl chains of the cell membrane. The anisotropy values of DPH-integrated intact cells after treatment with carvacrol and controls (Triton X-100 and Tween-20) are shown in Table 1. Different anisotropy values between the untreated, carvacrol and sanitizer-treated cells reflect their membrane viscosity. Our results show a significant decrease in the anisotropy value with increasing carvacrol-treated bacterial cells compared with the control untreated cells, suggesting enhanced viscosity/fluidity in the cell membrane of carvacrol-treated cells.

### 3.6. Carvacrol Preferentially Binds to P.G., P.E., and Partially to CL in the Bacterial Membrane

Preliminary investigations showed that the best solvent system for our TLC experiments is chloroform-methanol-acetic acid (65:25:10), and the ideal detection dye is alcoholic vanillin solution (Figure 5A,B). Carvacrol-treated P.G. and P.E. precipitates showed spots of their respective phospholipid, and their respective supernatants showed no unbound carvacrol spots. Therefore, the initially added carvacrol was assumed to be bound entirely with P.G. and P.E. However, both CL-treated carvacrol precipitant and supernatant showed CL spots, whereas its supernatant additionally showed an unbound carvacrol spot. Observation of CL samples suggested that carvacrol may be partially bound with CLs.

Two different spots of P.C. phospholipids were detected in the supernatant of the P.C.-treated sample with a spot similar to the initially added carvacrol. Therefore, we suggest that P.C. may break down by the effect of carvacrol but may not bind with degraded structures. Observation of no spots in Ch-treated precipitate and ChS and carvacrol spots similar to the initially added spots suggested that carvacrol may not bond with Ch. Furthermore, the respective precipitants of daptomycin-treated samples showed P.G., P.E., P.C., and CL spots and no trace of unbound daptomycin in the supernatant of these four phospholipids. Carvacrol and daptomycin left prominent P.C. fractions (two different spots) in the supernatants.

## 4. Discussion

Carvacrol is a phenolic monoterpenoid present primarily in essential oils of herbal plants, including thyme and oregano [1]. Several studies have reported the biological properties of carvacrol and suggested its potential clinical and natural health product applications [12,19,20]. We previously reported the rapid bactericidal activity of carvacrol against *S. pyogenes*. In this study, we noted, for the first time, the mode of action of carvacrol, which appears to target the membrane. Carvacrol alters membrane fluidity damaging the membrane integrity, and may interact with membrane phospholipids, P.G., P.E., and CL (Figure 6).

### 4.1. Carvacrol-Induced Cell Membrane Integrity Losses and Changes in Permeability, Potential, and Fluidity

We observed that carvacrol could permeabilize the cytoplasmic membrane and disrupt membrane potential. Membrane potential is the difference in electric potential across the membrane in intact bacterial cells [21]. The intact bacterial cell has a well-organized cytoplasm and membranes, which play an essential role in bacterial cytoskeletal spatial organization and cell division proteins [22]. Maintaining a constant proton gradient is crucial for bacterial cellular functions [23,24]. Loss of integrity, depolarization, and ion fluctuations in the membrane simultaneously affect its cellular functions, particularly ATP synthesis [25], active transport, cell division, and signal transductions [22].

We measured the membrane potential by the fluorescence intensity of molecular probes DiOC2(3). This molecule exhibits green fluorescence in bacterial cells, but the fluorescence shifts to red emission as the molecule self-associates at the higher cytosolic concentrations caused by higher membrane potential [26]. The red-to-green ratio in many Gram-positive bacteria is proportional to proton gradient intensity [27]. Interestingly, treatment with carvacrol caused a decrease in the red-to-green fluorescence ratio, suggesting that the mechanism of action involves disruption of membrane potential (depolarization of the membrane) in a concentration- and time-dependent manner.

Proton ionophores, such as CCCP, increase the proton permeability of the bacterial membrane, thereby dissipating membrane potential by eliminating the proton gradient [28]. Therefore, CCCP was used as a positive control in this study. In addition, *S. pyogenes* cells treated with CCCP showed depolarizing activity as expected in the early phase and recovered at the end of the incubation period. Our results agree with a previous study of CCCP on *S. pneumoniae* [29].

The results of the protoplast experiment confirmed that the ultimate target of carvacrol is the bacterial plasma membrane. TEM observations of protoplasts treated with carvacrol, such as uneven shape, intense cytoplasmic vacuolations, and cellular debris, revealed that carvacrol may have interacted with the lipid bilayer of *S. pyogenes* and induced structural changes. Furthermore, the observed nucleic acid leakage from the protoplast of *S. pyogenes* upon carvacrol treatments agreed with the results from the nucleic acid leakage pattern of intact bacterial cells upon similar treatments. Therefore, the cell membrane was the target of carvacrol against *S. pyogenes*. Similarly, other authors have suggested that the cytoplasmic membrane is the primary cellular target of essential oils containing carvacrol and thyme [30,31].

Carvacrol is a phenolic compound with a characteristic feature of the hydroxyl group on an aromatic ring. This phenolic hydroxyl group of carvacrol is crucial for its bactericidal activity. The importance of the phenolic structure of carvacrol and similar compounds, such as thymol, cymene, and menthol, in depolarizing the bacterial membrane was described against another Gram-positive pathogen, *Bacillus cereus* [32]. It was further suggested that destabilizing the cytoplasmic membrane by carvacrol acts as a proton exchanger, thus decreasing the pH gradient across the bacterial membrane. The resulting collapse of the proton motive force and depletion of the ATP pool eventually lead to cell death [32]. Therefore, we suggest that the hydroxyl group of carvacrol may contribute to destabilizing the cytoplasmic membrane phospholipids of *S. pyogenes*.

External environmental stress, such as temperature, drugs, and internal factors, such as the length and the degree of saturation of fatty acids tails, are a few factors that can influence bilayer fluidity of bacterial membranes [33,34]. Proper membrane fluidity is vital for maintaining the bacteria’s fundamental membrane barrier function. Decreasing membrane fluidity affects cell rigidity, leading to the malfunction of bacteria, whereas increasing fluidity causes cellular structural damage. Improper membrane fluidity interferes with essential bacterial cellular processes, such as maintaining membrane potential. Low membrane fluidity affects membrane leakiness in Gram-positive *B. subtilis*, which was reported, and indicates the weakening of membrane barrier functions and membrane homogeneity [35]. In contrast, it has been reported that the corresponding changes in the membrane fluidity had surprisingly little impact on membrane potential in *B. subtilis* [35]. Our previous findings agree with this pattern, where leakage of cytoplasmic biomolecules (proteins, DNA, and RNA) from the *S. pyogenes* membrane after carvacrol treatment was observed [7].

Fluorescence anisotropy results show a decrease in anisotropy values when cells are incubated with carvacrol, indicating an increase in membrane fluidity. The DPH probe was used in the experiment, and it is a highly hydrophobic molecule that inserts into the lipid core of the bacterial membrane [36]. DPH is oriented parallel to the axis of the lipid acyl chain, and its mode of motion is assumed to resemble the rotational diffusion of the lipid chains [36]. Therefore, membrane micro-viscosity/fluidity is explained as the rotational diffusion of the fatty acyl chains in the bacterial membrane interior (phospholipids) [36]. However, bacterial membrane phospholipids have different movements, such as lateral diffusion in the membrane plane and rotation around an axis perpendicular to the membrane plane [37]. Carvacrol caused an anisotropy decrease, indicating more significant alterations in configurations of the phospholipid bilayers, degree of lipid packing, and membrane thickness. Furthermore, microscopic observations of cell damages support the above results that carvacrol induced loss of bacterial membrane fluidity, which ultimately changed the cell shape.

### 4.2. Interaction between Carvacrol and Membrane Phospholipids

Bacteria contain different phospholipids (P.G., P.E., P.C., CL) in their cytoplasmic membrane in different proportions. Furthermore, the lipophilic monoterpenes have been reported to integrate with the membrane phospholipids and cause leakages [32,38]. Therefore, carvacrol, a lipophilic monoterpene, may also play a similar role in its antibacterial activity. However, the interaction between carvacrol and membrane phospholipids in *S. pyogenes* is not fully established. Therefore, it is interesting to investigate what types of membrane phospholipids of *S. pyogenes* are affected by the carvacrol treatment and suggest a potential mechanism of its interaction with the bactericidal activity of carvacrol.

The TLC experiments suggest that carvacrol does interact with specific phospholipids, such as P.E., P.G., and CL. A previous study showed that carvacrol, cinnamaldehyde, and geraniol could modify the lipid monolayers consisting of P.E., P.G., and CL by integrating into the phospholipid monolayer, forming aggregates of antimicrobial-lipid complexes, reducing the packing effectiveness of the acyl chains (tails) of phospholipids, increasing the membrane fluidity, and altering the total dipole moment in the monolayer bacterial cell membrane model [38].

Phospholipid acyl chains in the membrane lipids are connected through the van der Waals interactions [39]. Therefore, external disturbance to these non-covalent bonds causes fluidization of membrane lipids [39]. Similar fluidization monoterpenes were reported previously, namely eucalyptol, pulegone, terpineol, and thymol [40]. Furthermore, these compounds were observed to fluidize the liposomal membrane by interacting with the alkyl chains of P.C. liposomes [40].

Based on the present study, we suggest that carvacrol, an isomer to thymol, may also interact with the hydrophobic acyl chains of bacterial membrane phospholipids, creating a fluidizing effect on the lipid membrane (Figure 6). The expansion of lipid chains by carvacrol may destabilize the membrane and, consequently, the leakage of cytoplasmic content, as we reported here and previously [7]. Interference of carvacrol with bacterial membranes of *B. cereus* was previously reported as a mechanism of action of carvacrol and has shown that it changes the permeability of H^+^ and K^+^, obliterates essential functions, and ultimately causes cell death [32].

In general, the accumulation of hydrophobic phenolic compounds in hydrophobic phases, which occupy more than the usual space between fatty acyl chains, subsequently causes conformational changes in the phospholipid bilayer [41]. According to the observations, we suggested that carvacrol may accumulate in the membrane hydrophobic phases, affecting the intimate arrangement and stability of the phospholipid bilayer. Changes in the membrane phospholipid bilayer expand the membrane, as illustrated in Figure 6, which becomes more permeable and decreases membrane potential.

Daptomycin is a cyclic lipopeptide antibiotic with a broad spectrum of activity against Gram-positive bacteria, including *Streptococcus* spp. [42]. Although daptomycin mechanisms have not yet been precisely defined, it has been previously investigated and found to be due to the direct influence on the inhibition of biosynthesis of the cell membrane and/or cell wall component (including PGN, LTA) [43]. Daptomycin’s mechanisms against Gram-positive bacteria interact with the membrane P.G. [43,44]. Based on the findings of TLC experiments, we also suggest that daptomycin interacts with membrane P.G., P.E., P.C., and CL.

Carvacrol causes changes in cell membranes, such as depolarization, increased permeability, and leakage of the cytoplasmic contents, ultimately leading to bacterial cell death. Moreover, we found that carvacrol may preferentially target the membrane phospholipids of P.E., P.G., and partially CL. To our best understanding, this is the first report which shows that the primary target of bactericidal carvacrol is the membrane of *S. pyogenes*. Therefore, as a cell membrane-targeted novel antibacterial agent, with its plant origin, non or less toxic to human cells, and generally regarded as safe status, carvacrol is a potent natural agent for treating erythromycin-resistant *S. pyogenes* and drug-resistant pathogens.

## 5. Conclusions

We investigated the mechanism of action of carvacrol against *S. pyogenes* and its rapid bactericidal activities. The *S. pyogenes* membrane provides a target for carvacrol. According to our findings, carvacrol reduces the membrane potential of *S. pyogenes* cells while increasing membrane permeability and fluidity in a concentration-dependent manner. Furthermore, we propose that carvacrol’s bactericidal activity is mediated by preferential binding to negatively charged phosphatidylglycerol, cardiolipin, and zwitterionic phosphatidylethanolamine in bacterial cell lipid bilayers. These findings suggest that carvacrol is a possible candidate for developing novel natural health products, such as throat vapors, throat lozenges, or mouthwashes, for managing streptococcus pharyngitis.

## Figures and Tables

**Figure 1 pharmaceutics-14-01992-f001:**
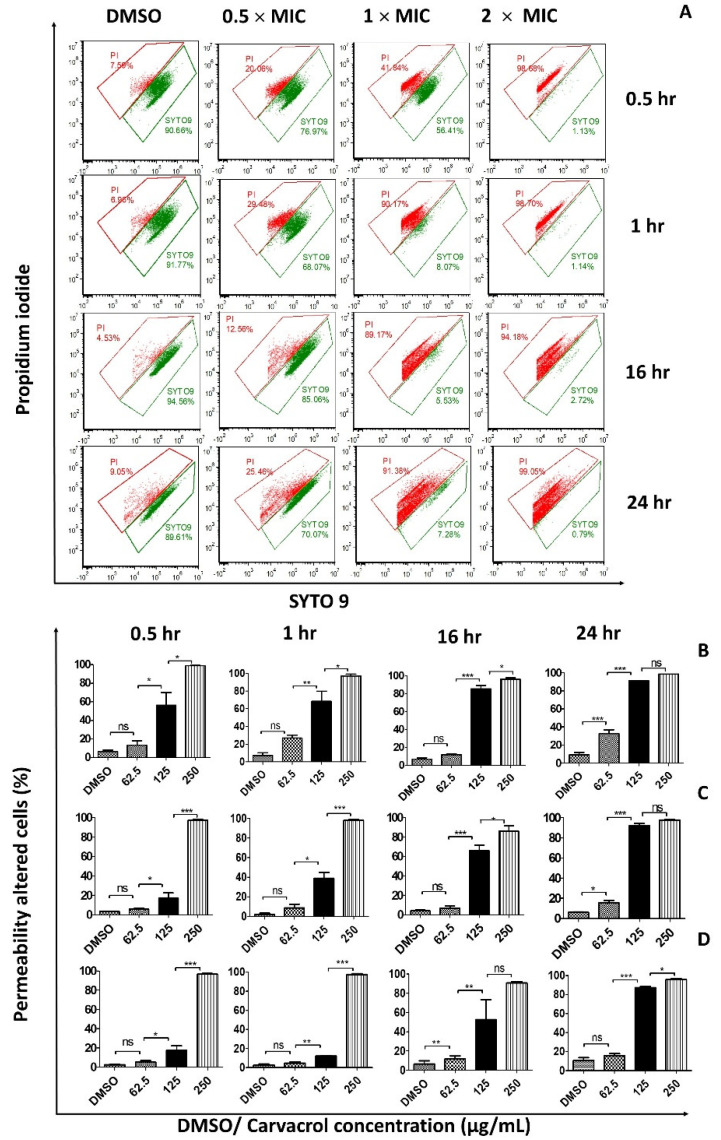
Carvacrol increases the permeability of the *Streptococcus pyogenes* bacterial cell membrane in a concentration- and time-dependent manner. Bacteria cells were treated with different carvacrol concentrations or vehicle control for 0.5, 1, 16, and 24 h, stained with SYTO9/PI and analyzed by flow cytometry. Red and green fluorescence intensity was measured using PC5.5 and FITC channels. Laser excitation/emission wavelengths of 485/542 nm for SYTO9 and 485/610 nm for PI were used. (**A**) Representative FACS cytograms of vehicle control (0.25% DMSO) or 0.5 × MIC, 1 × MIC, 2 × MIC of carvacrol-treated ATCC 19615 strain over four different incubation periods. The population of SYTO9-positive and PI-positive cells was shown in the green and red polygonal, respectively. The bar graphs show the cell permeability changes in (**B**) ATCC 19615, (**C**) clinical isolate, and (**D**) spy 1558 strains treated with different carvacrol concentrations over different incubation periods. Data shown are the percentages of membrane-altered cells (percentage of PI-positive cells) and expressed as mean ± SEM from three independent experiments. (Differences among means were compared using Tukey’s test; * *p* < 0.05, ** *p* < 0.01, and *** *p* < 0.001. MIC: Minimum inhibitory concentration = 125 µg/mL. PI: Propidium iodide.

**Figure 2 pharmaceutics-14-01992-f002:**
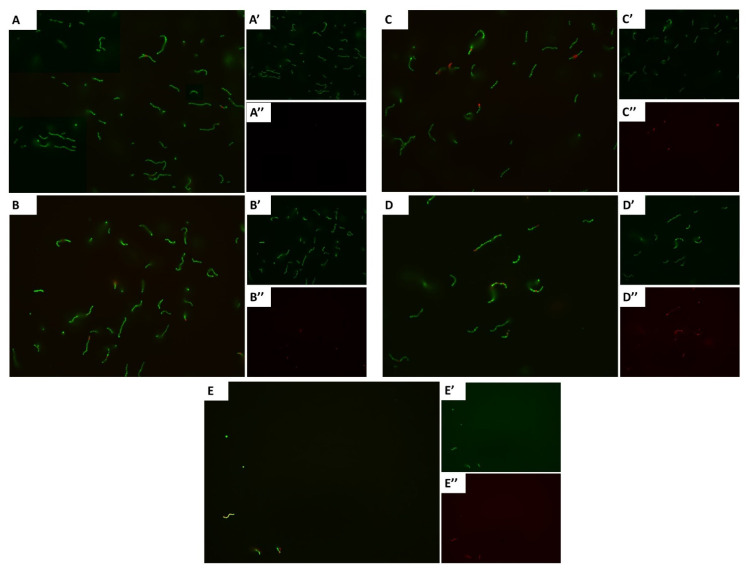
Bacterial viability is visualized in the fluorescence microscopy for *Streptococcus*
*pyogenes* cells. (**A**) Untreated control and treated with carvacrol at (**B**) 0.125 × MIC, (**C**) 0.25 × MIC, (**D**) 0.5 × MIC, and (**E**) 1 × MIC concentrations; MIC = 125 µg/mL. The cells are labeled with LIVE/DEAD BacLight^®^ bacterial-viability dyes, as STYO9 in green (‘) and propodeum iodide (PI) in red (‘’). Cells with a damaged/dead membrane show red color fluorescence, whereas cells with a viable and intact membrane show green fluorescence.

**Figure 3 pharmaceutics-14-01992-f003:**
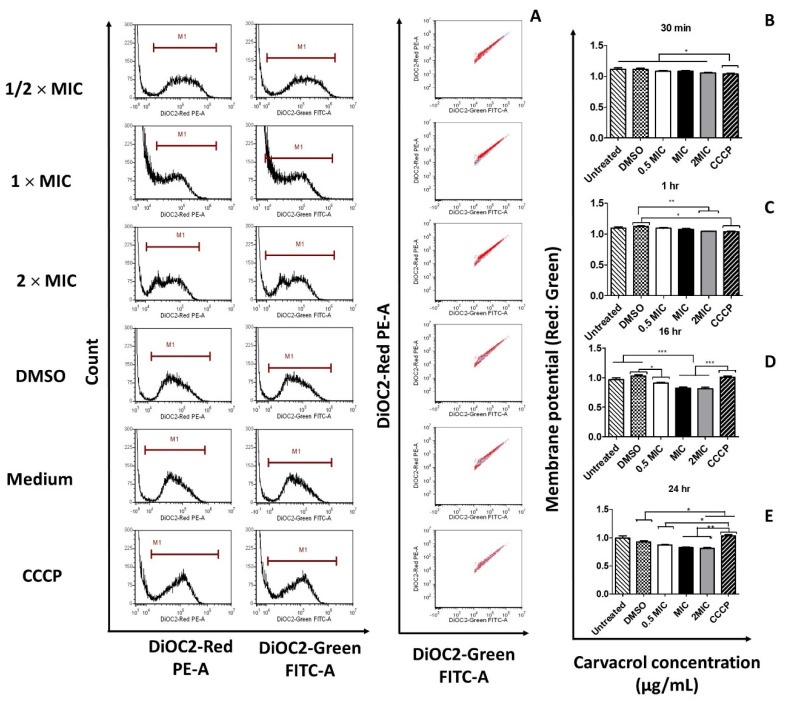
Effect of carvacrol treatments on the potential membrane changes in *Streptococcus pyogenes*. Bacterial suspensions were treated with 2 × MIC, 1 × MIC, and 1/2 × MIC of carvacrol, or DMSO vehicle or 5 μM of carbonyl cyanide 3-chlorophenylhydrazone (CCCP), then incubated with 300 μM DiSO2(3) for 24 h; MIC = 125 μg/mL. The fluorescence intensity of red and green was measured using FITC and PI channels in a FACS Cyto FLEX flow cytometer. The intensity of red (left panel), green fluorescence ((**A**), middle panel), and cells exhibiting both fluorescence intensities (right panel) are shown. Red dots represent CCCP, and blue dots represent carvacrol treatments in cytometry profiles (right panel). Representative fluorescence panels were shown for the *S. pyogenes* strain of ATCC 19615 suspension treated with carvacrol for 24 h. The bar graph shows the membrane potential without (untreated control) or with DMSO vehicle or CCCP or different concentrations of carvacrol treatment for different incubation periods of (**B**) 30 min, (**C**) 1 h, (**D**) 16 h, and (**E**) 24 h. Flow cytometer data were collected with *log* amplifications, and red-to-green ratios were calculated using population mean fluorescence intensities (MFI) of carvacrol treatments in the presence or absence of CCCP. Results are expressed as the red-to-green ratio of MFI ± SEM of three independent experiments. Statistical analysis was performed using one-way ANOVA, and the differences among means were compared using Tukey’s test; * *p* < 0.05, ** *p* < 0.01, and *** *p* < 0.001.

**Figure 4 pharmaceutics-14-01992-f004:**
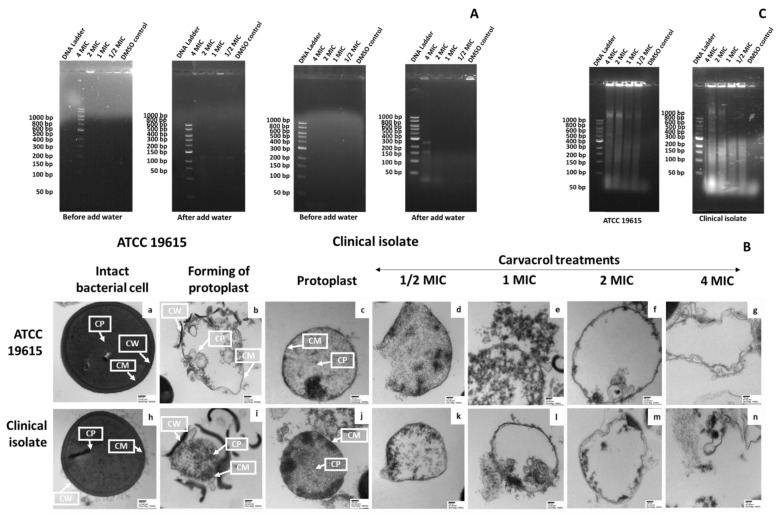
Confirmation of *Streptococcus pyogenes* protoplast formation, carvacrol effect on membrane damage of *S.*
*pyogenes* protoplast, and cytoplasmic nucleic acid leakage. (**A**) Agarose gel (0.8%, *w*/*v*) electrophoresis and gel red staining of centrifuged clinical isolate-protoplast pellets after formation and busting out by adding water. (**B**) Transmission electron micrographs of thin sections of intact bacteria without treatment (a,h), a representative example of a progressive stage of protoplast of the effect of mutanolysin in protoplast buffer (15 min at 37 °C) (b,i), protoplast resulting from exposure in protoplast buffer with mutanolysin (30 min at 37 °C), and ATCC 19615 and clinical isolate protoplast cell suspensions (OD_600_ = 0.6) treated with DMSO (c,j) or 1/2 × MIC (d,k), 1 × MIC (e,l), 2 × MIC (f,m), and 4 × MIC (g,n) of carvacrol concentrations for 2 h at 37 °C. Magnifications and bars: × 50,000 (200 nm) and × 100,000 (100 nm). CW: Cell wall; CM: Cytoplasmic membrane; CP: Cytoplasm. (**C**) Agarose gel (0.8%, *w*/*v*) electrophoresis, gel red staining of ATCC 19615, and clinical isolate-protoplast supernatants after exposure to the different carvacrol concentrations; 1 kb ladder as reference.

**Figure 5 pharmaceutics-14-01992-f005:**
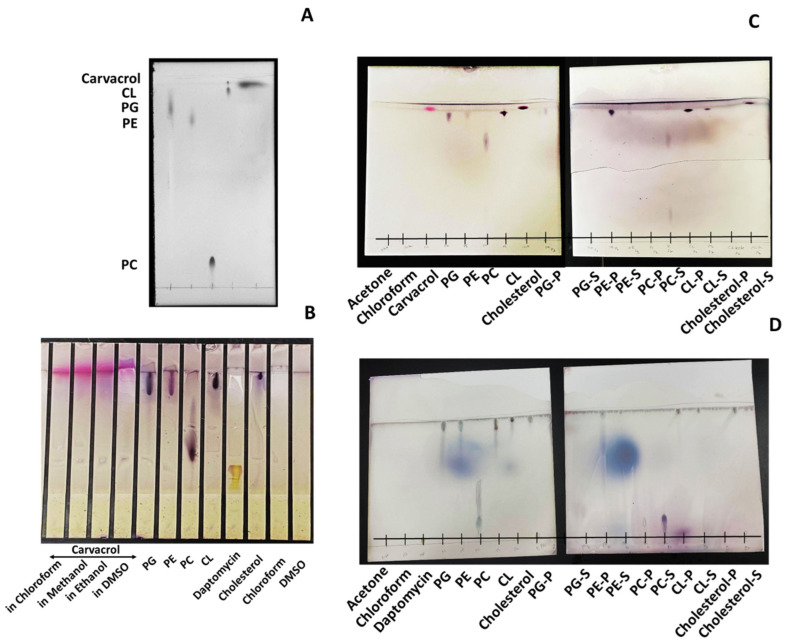
Visualization of thin-layer chromatography (TLC). (**A**) UV visualization of phospholipid standards (25 mg/mL, each) and carvacrol (12.5 mg/mL), (**B**) selection of better solvents and detection method for all samples, (**C**) carvacrol, and (**D**) daptomycin interaction with phospholipids. Phospholipids: carvacrol/daptomycin (2:1 *v*/*v*, in chloroform) were incubated for 1 h, and the phospholipids were precipitated using cold acetone. Precipitant (P) and supernatant (S) were collected by high-speed centrifugation (21,000× *g*), drying, and reconstituted with chloroform. The TLC plates were developed with the solvent system of chloroform-methanol-water (65:25:10 *v/v/v*). TLC plates were then stained with alcoholic vanillin–sulphuric acid solution, heated in an oven, and visualized under UV. The colors of the spots were compared and identified in photographs of TLC plates, and their mobilities (Rf) were compared with those of phospholipids. P.G.: Phosphatidylglycerol; P.E.: Phosphatidylethanolamine; P.C.: Phosphatidylcholine; CL: Cardiolipin; Ch: Cholesterol.

**Figure 6 pharmaceutics-14-01992-f006:**
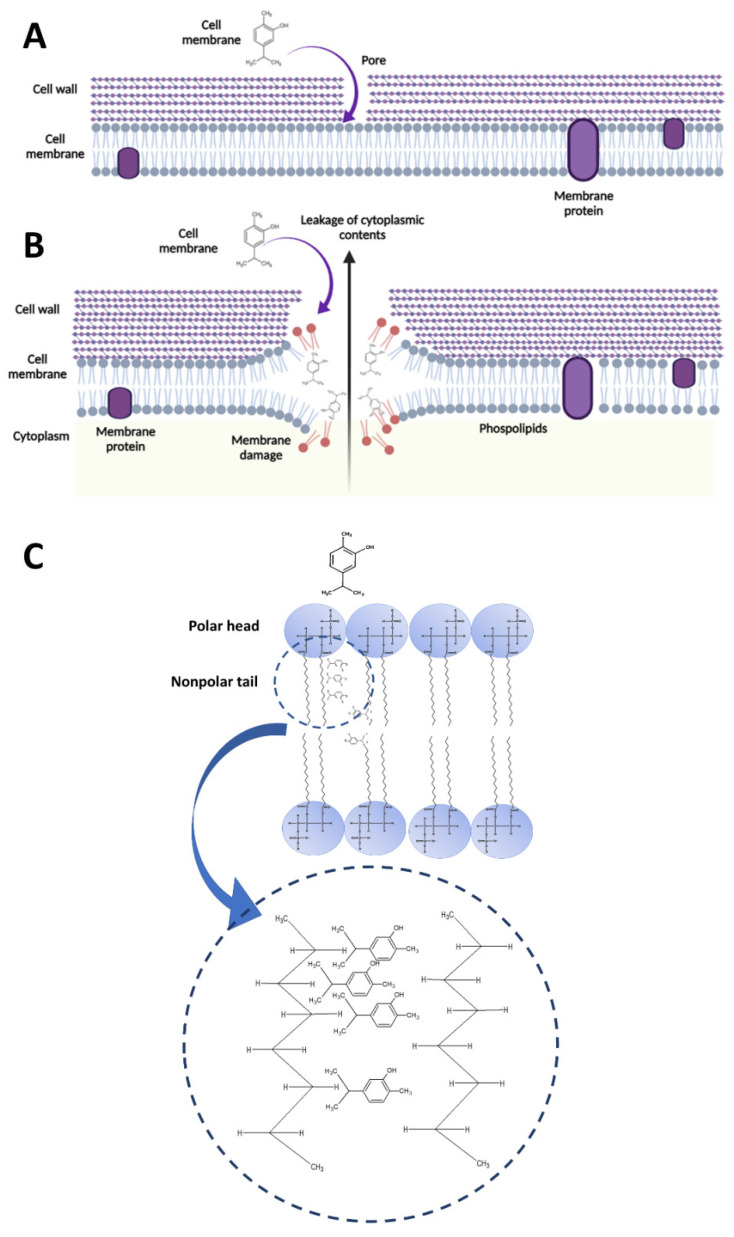
Potential mechanism of action of carvacrol in damaging the cell membranes of *Streptococcus*
*pyogenes*. (**A**) Due to its hydrophobic nature, carvacrol penetrates through cell wall pores to the cell membrane and (**B**) interacts with the bacterial cytoplasmic membrane lipid bilayer. Carvacrol may align between fatty acid chains of the membrane phospholipids, specifically P.G., P.E., and CL, causing the expansion and destabilization of the membrane structure by increasing the fluidity and permeability. As a result, cytoplasmic cell contents will be leaking from the cytoplasm, which ultimately causes cell death. (**C**) The accumulation of carvacrol in hydrophobic phases, which occupies more than the usual space between fatty acyl chains, subsequently causes some conformational changes in the phospholipid bilayer. P.G.: Phosphatidylglycerol; P.E.: Phosphatidylethanolamine; CL: Cardiolipin.

**Table 1 pharmaceutics-14-01992-t001:** Changes in membrane fluidity in the presence of carvacrol on *Streptococcus pyogenes* spy 1558 live cells.

Samples	DPH Anisotropy Value *	
	1 h	24 h
PBS	0.012 ± 0.01	0.011 ± 0.01
DMSO	0.012 ± 0.01	0.023 ± 0.01
Carvacrol (1 × MIC)	0.012 ± 0.01	0.018 ± 0.01
Carvacrol (2 × MIC)	0.005 ± 0.00	0.047 ± 0.00
Triton X-100	0.009 ± 0.01	0.042 ± 0.00
Tween-20	0.030 ± 0.00	0.034 ± 0.00

* DPH anisotropy values represent the mean ± standard error. DPH: 1,6 Diphenyl 1,3,5 hexatriene; MIC: Minimum inhibitory concentration (125 µg/mL); DMSO: Dimethyl sulfoxide; PBS: Phosphate-buffered saline.

## Data Availability

All the data presented in the paper.

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
