# Peer review of "Bactericidal Activity of Carvacrol against *Streptococcus pyogenes* Involves Alteration of Membrane Fluidity and Integrity through Interaction with Membrane Phospholipids"

_pharmaceutics, 2022, doi:10.3390/pharmaceutics14101992_

Round 1

Reviewer 1 Report

The present manuscript investigated the bactericidal mechanism of carvacrol using three strains of S. pyogenes. The authors performed Flow cytometry experiments determine carvacrol's membrane permeabilization and cytoplasmic membrane depolarization activity. S. pyogenes protoplast model was used to investigate the effects of carvacrol on ultrastructural morphological changes of the membrane using transmission electron microscopy. The fluidity of the cell membrane was measured by steady-state fluorescence anisotropy. Thin layer chromatographic (TLC) profiling was conducted to study the affinity of carvacrol for membrane phospholipids.

The same research group published a very similar study previously (Wijesundara NM, Lee SF, Cheng Z, Davidson R, Rupasinghe HPV. Carvacrol exhibits rapid bactericidal activity against Streptococcus pyogenes through cell membrane damage. Sci Rep. 2021 Jan 15;11(1):1487. doi: 10.1038/s41598-020-79713-0. PMID: 33452275; PMCID: PMC7811018.)

Why did the authors not publish the results together? The question answered by the present manuscript is unclear, and the authors need to emphasize the novelty of the current manuscript. In this way, the abstract's conclusion and the manuscript's conclusion should focus on the mechanisms of action and not on the action. For example, the authors conclude the abstract as follows: “Carvacrol exhibited rapid bactericidal action against S. pyogenes by disrupting the bacterial membrane and increasing permeability.” This is very similar to the abstract of the previous paper cited above as follows: “We also demonstrated the potential mechanism of action of carvacrol through compromising the cell membrane integrity. Carvacrol induced membrane integrity changes leading to leakage of cytoplasmic content such as lactate dehydrogenase enzymes and nucleic acids. We further confirmed dose-dependent rupturing of cells and cell deaths using transmission electron microscopy.

The authors also used the same methodology, transmission electron microscopy.

Besides it, the introduction section presents the objective of the manuscript twice. Please, keep just one mention to the aim.

Please, remove the objective from the conclusion and focus it on the novelty of the present manuscript.

At last, bacterial should be written in italics.

Author Response

Reviewers Response:

  • As suggested, we have emphasized the novel findings of the mechanism of action in both the abstract and conclusion sections of the improved manuscript.
  • We have removed the repetition of the objective in the manuscript. Thanks.
  • We have made sure that the bacterial names are in italics. We had all the scientific names in italic in our original manuscript but during the formatting to Journal format, they have got changed.

Reviewer 2 Report

The authors describe the antimicrobial mechanism of (free) carvacrol on the Gram-positive bacterium S. pyogenes using several experimental approaches. They found that carvacrol specifically targets the cytoplasmic membrane, interacting with specific phospholipids and modifying its properties, structure and biological functions, up to lysis and cell death.

I found the work interesting, however I think there are small changes to be made for its publication. Below are the main comments/remarks.

Section 3.4.

Line 333. From figure 4A in my view, a dose-dependent effect of carvacrol could only be evident for the clinical isolate. How did you quantize the nucleic acids released by the lysis of protoplasts in figure 4A?

In figure 4A did you analyze the “TCA perceptible nucleic acid released into supernatant” of the protoplasts treated with carvacrol (with the method reported at lines 190-198), or the nucleic acids released from the pellet of the protoplasts treated with carvacrol after osmotic shock with water ( with the method reported at lines 185-188)? In the latter condition, I aspect a different results from the one presented in figure 4A.

Fig. 4B.  Indicate each micrograpy with letters o numbers and make visible CW, CP, CM.

In conclusion, the section described in fig. 4 is confusing and I think it is necessary to better clarify the aspects reported above in the caption and paragraphs 2.6.2, 2.6.3 (methods) and 3.4.1 (results). Also, better detail the differences between Figure 4A and Figure 4C.

Section 3.6.

To accept the deductions made by the TLC analysis it is necessary to better present the results, add further clarifications and improve the figures. Follow below.

From the results of TLC I deduced that the carvacrol-phospholipid association determines the precipitation of the complex in cold acetone and that unbound phospholipids and freee carvacrol do not precipitate. It's right? Explain the correct principle on which the assay is based in the results (or methods) and, if possible, also include some references. Similarly for daptomycin.

Possibly, use better quality images for Figures 5 A and B, 5B is very bad. I find Figure S3 clearer than Figure 5, although the resolution of the spots is lower. Try to improve the images also by using the arrows to highlight the main spots, those difficult to view and/or decisive for the discussion of the data.

Line 381. From the figure of TLC in my view (both, fig. 5 and fig.S3) the spot of the carvacrol is not evident in the lane CL-S.

Lines 395-396. I see this profile only in figure S3, but not in figure 5.

Lines 398-399: “……and Ch and carvacrol spots similar to initially added spots….”. Do you mean in Ch-S?

Section 4 and 5 (discussion and conclusion).

The discussion is confusing in some parts and is very redundant, especially from lines 543 to 563. It should be better written, eliminating all repetitions. Please, also consider the comments below.

Lines 436-446. Here you discuss the principle of the method used. It would be useful eliminate this part from section 4 and to include it in the "Methods" or "Results", in order to help the reader to interpret and understand your data while reading the paper.

Lines 468-469. Give more information on the reasons for this suggestion and if possible, include some references on the role of the hydroxyl group of carvacrol.

Lines 485-490. As for lines 436-446.

Latest remarks

Check the punctuation and the use of italics for the name of the bacterial species

Author Response

Reviewer 2:

The authors describe the antimicrobial mechanism of (free) carvacrol on the Gram-positive bacterium S. pyogenes using several experimental approaches. They found that carvacrol specifically targets the cytoplasmic membrane, interacting with specific phospholipids and modifying its properties, structure and biological functions, up to lysis and cell death.

I found the work interesting, however, I think there are small changes to be made for its publication. Below are the main comments/remarks.

Section 3.4.

Line 333. From figure 4A in my view, a dose-dependent effect of carvacrol could only be evident for the clinical isolate. How did you quantize the nucleic acids released by the lysis of protoplasts in figure 4A?

Reviewers Response:

Yes, we agree. It is difficult to demonstrate the dose-dependent effect using the gel electrophoresis. However, our primary goal was to show whether protoplasts were properly formed or not. Therefore, we have not attempted to how much nucleic acid leaked and not quantified. The leakage of nucleic acid can be observed once the osmotic pressure is removed. In summary, the experiment was conducted to investigate whether burst open cells release nucleic acids (qualitative) after removing osmotic support.

In figure 4A did you analyze the “TCA perceptible nucleic acid released into supernatant” of the protoplasts treated with carvacrol (with the method reported at lines 190-198), or the nucleic acids released from the pellet of the protoplasts treated with carvacrol after osmotic shock with water ( with the method reported at lines 185-188)? In the latter condition, I aspect a different results from the one presented in figure 4A.

Reviewers Response:

In figure 4A, we are presenting the nucleic acids released from the pellet of the protoplasts (without any treatment) before and after osmotic shock with water. 4A confirms the proper formation of protoplasts.

Fig. 4B.  Indicate each micrograpy with letters o numbers and make visible CW, CP, CM. Also, arrows of Fig 4 V need to be fixed?

Reviewers Response:

As suggested, we have rearranged the Figure 4.

Letters were included in each micrography; Enlarged the CW, CP, and CM letters, boxes, and arrows; The thickness of the arrows was adjusted. Also, the arrows of Fig 4 V have been fixed.

In conclusion, the section described in fig. 4 is confusing and I think it is necessary to better clarify the aspects reported above in the caption and paragraphs 2.6.2, 2.6.3 (methods) and 3.4.1 (results). Also, better detail the differences between Figure 4A and Figure 4C.

Reviewers Response:

Here is an explanation of Figure 4A versus 4C:

Figure 4A: We analyzed the nucleic acids released from the pellet of the protoplasts (without any treatment) before and after osmotic shock with water. Thus, Figure 4A confirms the proper formation of protoplasts.

Figure 4C: We analyzed the nucleic acids released from the pellet of the protoplasts after challenging with different concentrations of carvacrol.

Section 3.6.

To accept the deductions made by the TLC analysis it is necessary to better present the results, add further clarifications and improve the figures. Follow below.

From the results of TLC I deduced that the carvacrol-phospholipid association determines the precipitation of the complex in cold acetone and that unbound phospholipids and freee carvacrol do not precipitate. It's right? Explain the correct principle on which the assay is based in the results (or methods) and, if possible, also include some references. Similarly for daptomycin.

Reviewers Response:

As suggested, Figure 5 is improved in the revised manuscript.

As far as our up-to-date search of scientific data bases, no previous references are found for this nature of experiment. Therefore, we designed the experiment and used TLC to generate a qualitative idea of whether carvacrol binding ability could be detectable or not. Sorry, there is no previous method/references to include.

Possibly, use better quality images for Figures 5 A and B, 5B is very bad. I find Figure S3 clearer than Figure 5, although the resolution of the spots is lower. Try to improve the images also by using the arrows to highlight the main spots, those difficult to view and/or decisive for the discussion of the data.

Line 381. From the figure of TLC in my view (both, fig. 5 and fig.S3) the spot of the carvacrol is not evident in the lane CL-S.

Lines 395-396. I see this profile only in figure S3, but not in figure 5.

Lines 398-399: “……and Ch and carvacrol spots similar to initially added spots….”. Do you mean in Ch-S?

Reviewers Response:

  • As suggested, we have improved the graphical quality of Figure 5.
  • Supplementary Figure S3 was removed, since the key content of Fig S3 is now combined with Figure 5.
  • Thanks, Ch is changed to Ch-S.

Section 4 and 5 (discussion and conclusion).

The discussion is confusing in some parts and is very redundant, especially from lines 543 to 563. It should be better written, eliminating all repetitions. Please, also consider the comments below.

Reviewers Response:

As suggested, the discussion section has been improved to avoid redundance.

Lines 436-446. Here you discuss the principle of the method used. It would be useful eliminate this part from section 4 and to include it in the "Methods" or "Results", in order to help the reader to interpret and understand your data while reading the paper.

Lines 468-469. Give more information on the reasons for this suggestion and if possible, include some references on the role of the hydroxyl group of carvacrol.

Reviewers Response:

  • The suggested modification was made.
  • As suggested, a new reference is included.

Ultee, A.; Bennik, M.H.J.; Moezelaar, R. The phenolic hydroxyl group of carvacrol is essential for action against the food-borne pathogen Bacillus cereus. Appl Environ Microbiol 2002, 68, 1561-1568, doi:10.1128/aem.68.4.1561-1568.2002.

Latest remarks

Check the punctuation and the use of italics for the name of the bacterial species

Reviewers Response:

Thanks, the errors/typos created due to formatting is now fixed.

Reviewer 3 Report

There are many articles describing the bactericidal activity of phenols as natural alternatives to antibiotic-resistant microorganisms. However, very little is focused on how those act on bacterial cells. Their mechanism of action is not yet well understood. Therefore, in my opinion, the presented work is valuable and will make a significant contribution to the development of science. However, I have minor comments.
I will not judge the English language - I am not a native. However, in my opinion, the publication is written comprehensively.
Please put the MIC concentrations in the first place of their occurrence. The authors have included the MIC concentrations but later in the work, which is a bit confusing for the reader.
 Check all the work - the Latin name of the bacterial species should be in italics. Also, make sure you have the correct indexes for chemical names.

Author Response

Reviewer 3:

There are many articles describing the bactericidal activity of phenols as natural alternatives to antibiotic-resistant microorganisms. However, very little is focused on how those act on bacterial cells. Their mechanism of action is not yet well understood. Therefore, in my opinion, the presented work is valuable and will make a significant contribution to the development of science. However, I have minor comments.

I will not judge the English language - I am not a native. However, in my opinion, the publication is written comprehensively.

Please put the MIC concentrations in the first place of their occurrence. The authors have included the MIC concentrations but later in the work, which is a bit confusing for the reader.

Reviewers Response:

Thank you. MIC concentration is included in the earlier sections of the manuscript (page 3): “According to our previous finding,  carvacrol showed effectiveness against an erythromycin-resistant strain of S. pyogenes, spy 1558 at a minimum inhibitory concentration (MIC) of 125 µg/mL”

Check all the work - the Latin name of the bacterial species should be in italics. Also, make sure you have the correct indexes for chemical names.

Reviewers Response:

All the scientific names are in italics. This error occurred during file formatting to MDPI style.

Reviewer 4 Report

First and foremost, the paper is schematically and nicely written. 

It is a topic with high relevance in the scientific community and the use of terpenoids in different fields is becoming more popular, since thyme, rosemary, oregano, and other herbs have shown outstanding antibacterial activities as well as antifungal, insect repellants among others. 

As far as I am concerned, the paper should be accepted; nonetheless, there are some remarks I consider should be addressed: 

1.  I know is difficult but avoid using the verb kill if it is possible, instead, and if the sentence allows it, use inhibit counteract or verbs like that. 

2. Avoid using first person plural, instead use impersonals

3.  Line 514. '' Phospholipid acyl chains in the membrane lipids are connected through the “van der Waals” interactions [39].''

I do not understand why the authors use quotation marks to refer to the van der Waals interactions. It should be better explained. 

Author Response

Reviewers Response:

  1. We have replaced “kill” with “inhibit”, where appropriate.
  2. Impersonal style has been used.
  3. We have removed the quotation marks used for “van der Waals” interactions.

Round 2

Reviewer 1 Report

I am still not convinced about the novelty of the present manuscript. The authors did not try to explain the novelty of the manuscript. They only modified the conclusion and included that carvacrol has an affinity with specific membrane phospholipids such as P.E., P.G., and CL. 

This information did not justify another publication with the same theme.